# Assessing the Generalization of Machine Learning-Based Slope Failure Prediction to New Geographic Extents

Aaron E. Maxwell [1],*[iD], Maneesh Sharma [2], J. Steven Kite [1], Kurt A. Donaldson [2], Shannon M. Maynard [2] and Caleb M. Malay [2]

[1]  Department of Geology and Geography, West Virginia University, Morgantown, WV 26505, USA; steve.kite@mail.wvu.edu
[2]  West Virginia GIS Technical Center, Morgantown, WV 26505, USA; Maneesh.Sharma@mail.wvu.edu (M.S.); Kurt.Donaldson@mail.wvu.edu (K.A.D.); smmaynard@mail.wvu.edu (S.M.M.); cmmalay@mix.wvu.edu (C.M.M.)
*  Correspondence: Aaron.Maxwell@mail.wvu.edu; Tel.: +1-304-293-2026

**Abstract:** Slope failure probabilistic models generated using random forest (RF) machine learning (ML), manually interpreted incident points, and light detection and ranging (LiDAR) digital terrain variables are assessed for predicting and generalizing to new geographic extents. Specifically, models for four Major Land Resource Areas (MLRAs) in the state of West Virginia in the United States (US) were created. All region-specific models were then used to predict withheld validation data within all four MLRAs. For all validation datasets, the model trained using data from the same MLRA provided the highest reported overall accuracy (OA), Kappa statistic, F1 Score, area under the receiver operating characteristic curve (AUC ROC), and area under the precision-recall curve (AUC PR). However, the model from the same MLRA as the validation dataset did not always provide the highest precision, recall, and/or specificity, suggesting that models extrapolated to new geographic extents tend to either overpredict or underpredict the land area of slope failure occurrence whereas they offer a better balance between omission and commission error within the region in which they were trained. This study highlights the value of developing region-specific inventories, models, and high resolution and detailed digital elevation data, since models may not generalize well to new geographic extents, potentially resulting from spatial heterogeneity in landscape and/or slope failure characteristics.

**Keywords:** slope failures; landslides; light detection and ranging; LiDAR; digital terrain analysis; machine learning; random forest; spatial predictive modeling; generalization

## 1. Introduction

Slope failures, such as landslides, are a geohazard of global concern that often result in damage to personal property and public infrastructure, exacerbation of social and economic issues impacting already strained communities and governments, and loss of life [1–3]. Remote sensing, spatial predictive modeling, and machine learning (ML) techniques have proven valuable for inventorying or mapping existing slope failures (e.g., [4–9]) and predicting the likelihood or probability of slope failure occurrence or susceptibility (e.g., [10–17]). High spatial resolution digital terrain data, such as those derived using light detection and ranging (LiDAR), have improved our ability to identify slope failures on the landscape surface, even when obscured by vegetative cover, and generate detailed terrain variables to aid in modeling [17–22]. Further, such data are becoming more widely available; for example, the United States Geological Survey (USGS) is currently coordinating the collection of LiDAR data for the entire contiguous United States (US) via the 3D Elevation Program (3DEP) [23,24].

Despite these advancements, there is still a need to further refine and develop computational methods for generating map products and predictive models consistently over

large spatial extents and to make full use of slope failure inventories and digital terrain data. In this study, we specifically focus on the production of probabilistic slope failure occurrence models using LiDAR-derived terrain variables and manually digitized failure locations to explore the key issue of how well models trained in different landscapes, defined by Major Land Resource Areas (MLRAs), extrapolate to other MLRAs in the state of West Virginia in the US. In this study, we define slope failures as the movement of a mass of rock, earth, or debris down a slope [3,25].

This study is part of a larger project to develop a multi-hazard risk assessment for the entire state of West Virginia. For the slope failure component of the study, we are working on completing a statewide slope failure occurrence probability model along with a dataset of manually interpreted occurrence points, data which are made available through the web-based WV Landslide Tool (https://www.mapwv.gov/landslide) (accessed on 2 May 2021). Our prior study specifically focused on results for the Northern Appalachian Ridges and Valleys (NARV) MLRA, which constitutes the eastern portion of the state. This prior study concluded that including additional landscape variables, such as distance to roads and streams, soil properties, and bedrock geology characteristics, may not improve model performance in comparison to just using digital terrain variables derived from LiDAR. We specifically documented the value of the following topographic variables: slope gradient (Slp), surface area ratio (SAR), cross-sectional curvature (CSC), surface relief ratio (SRR), and plan curvature (PlC). We also noted the value of a large, quality training dataset and the robustness of the random forest (RF) ML algorithm to a large feature space, as model performance did not improve with variable selection [17].

This current study expands upon the prior study by focusing on an investigation of model performance when extrapolating to new geographic extents with disparate geomorphic, land use, disturbance, and/or geologic characteristics. Given that it is commonly very labor-intensive and time-consuming to develop new training data and models for new geographic extents, investigating the geographic generalization of region-specific models is a valuable research contribution, especially considering the increasing availability of detailed digital terrain data.

## 2. Background

### 2.1. Model Generalization

The generalization of spatial predictive models to new geographic extents has been investigated in prior studies (e.g., [26–30]). For example, Maxwell et al. [28] explored the generalization of palustrine wetland probabilistic occurrence models when trained in one physiographic region and applied to new regions, all within the state of West Virginia. They noted a reduction in performance, as measured with the area under the receiver operating characteristic curve (AUC ROC), and cautioned against extrapolating such predictive models to new geographic extents. Many recent studies have focused on the generalization of deep learning (DL) models specifically, which have been argued to generalize well to new data given the high level of data abstraction applied [26,30].

Many studies have documented successful DL model generalization to new, disparate data and/or other geographic extents [31,32]. For example, Maggiori et al. [33] reported accurate generalization of convolutional neural network (CNN)-based DL building detection models when trained in a subset of cities and applied to different cities. However, several studies have noted a reduced DL model performance when applied to new geographic extents or disparate data; for example, Maxwell et al. [29] quantified reduced model performance when a semantic segmentation DL model for extracting historic surface mine extents from topographic maps trained on data in eastern Kentucky was applied to new topographic maps in mining regions of southwestern Virginia and eastern Ohio. Similarly, Maxwell et al. [27] quantified reduced performance when an instance segmentation DL model for extracting valley fill faces, geomorphic features resulting from surface mine reclamation, was trained in West Virginia then applied to new data and geographic extents in Kentucky and Virginia. The model, which was trained using LiDAR-derived

data, performed especially poorly when used to predict disparate photogrammetrically-derived data.

We argue that the generalization performance of probabilistic spatial models generated using ML or DL techniques is problem- and method-specific. Thus, if there is a need to apply models to new geographic extents, then this generalization performance must be assessed. Or, validation of a model using data collected within the same geographic extent to which it was trained does not offer a valid assessment of model generalization performance to new geographic extents.

### 2.2. Probabilistic Spatial Models with Random Forest

This study makes use of the random forest (RF) [34] nonparametric, ML algorithm to generate probabilistic spatial models of slope failure occurrence. Single decision trees (DT) rely on recursive binary splits of the data to generate decision rules to partition the data into more homogeneous subsets. It has been shown that using an ensemble of decision trees, as opposed to a single tree, can improve model performance [34]. RF offers one means to accomplish this. Each tree in the RF model receives a subset of the available training data, selected using bootstrap sampling. Also, only a random subset of the available predictor variables is available for selecting a splitting rule at each node. The goal of limiting the training data available in each tree and the predictor variables available at each node is to reduce the correlation between the trees in the ensemble. Although each tree may generate a weaker prediction, the ensemble of weak predictors can be collectively robust due to reduced correlation and overfitting [17,34–37].

RF has shown many positive attributes for application to remotely sensed and geospatial data for predictive modeling tasks. First, it has been reported to be robust to complex and large feature spaces and can accept continuous and categorical predictor variables with varying scales and distributions [17,35,37–39], minimizing the need for feature selection or reduction. Because each tree in the ensemble does not use all the training samples, the withheld, or out-of-bag (OOB), samples can be used to assess model performance as long as the training samples are representative of the population and unbiased [34,36,37,39]. It also allows for the estimate of predictor variable importance within the model [34,37], which improves the interpretability of the model and allows for the identification and documentation of key explanatory variables.

The RF algorithm has been applied to a variety of spatial probabilistic mapping problems. For example, Evans and Cushman [40] applied RF to predict the probability of occurrence of conifer species using topographic, climate, and spectral variables. Strager et al. [41] applied the algorithm to predict the future expansion of surface coal mining in the Appalachian region based on coal seam and landscape characteristics. Maxwell et al. [28] and Wright and Gallant [42] both explored probabilistic wetland mapping.

This algorithm has specifically been applied to slope failure susceptibility and occurrence predictive modeling (e.g., [5,12,14,17,43–49]). Goetz et al. [44] assessed multiple models for susceptibility mapping including RF, generalized additive models (GAMs), weights of evidence (WoE), support vector machines (SVM), and bootstrap aggregated classification trees with penalized discriminant analysis (BPLDA). Their study concluded that RF provided the strongest predictive performance based on AUC ROC and true positive rate (TPR). However, the performance was not always statistically significantly better than that provided by all other tested algorithms. Youssef et al. [47] also documented strong performance for RF in comparison to other methods. Pourghasemi and Kerle [49] combined RF and evidential belief function (EBF) approaches for susceptibility modeling while Taalab et al. [14] documented the value of RF for making slope susceptibility predictions over large spatial extents. Catani et al. [43] undertook a systematic study of RF for slope failure susceptibility mapping to assess sensitivity and scaling issues relating to model hyperparameter settings, feature space and optimal predictor variable sets, spatial resolution of input variables, and training data characteristics. They concluded that the optimal settings and inputs vary based on the scale and resolution of the output model

and that optimal variables to undertake a prediction depend on algorithm settings and available data. This study highlights the need to assess model generalization to new geographic extents given variability in model performance when investigated using a systematic sensitivity analysis.

### 2.3. Digital Terrain Data for Landslide Mapping and Modeling

Generally, topographic variables derived from digital terrain models (DTMs) have been shown to be of value for predicting slope failure susceptibility or occurrence (e.g., [17,18,43,44,50]). In contrast, multispectral remotely sensed data are more commonly applied to slope failure mapping and detection problems and rely on unique spectral or textural signatures of the disturbed surface in comparison to other land cover or landscape features in the mapped extent [4–9,51–55]. The high spatial resolution digital terrain representations made available by LiDAR have been shown to be of particular value for landslide susceptibility or occurrence modeling [17,18,20,21,56]. Such models can offer enough detail for trained geomorphologists and geohazard professionals to identify slope failures and associated features, such as scarps and head and toe components [17,20,21]. LiDAR is an active remote sensing technique that relies on laser range finding, global navigation satellite systems (GNSS), and inertial measurement units (IMUs) to generate three-dimensional point clouds referenced to a geospatial datum and projection to represent landscapes with a high level of spatial and textural detail. Further processing of the point cloud data allows for differentiation of ground returns. These ground points can then be used to generate raster-based, continuous DTMs [20,21,57].

From LiDAR-derived DTMs a variety of topographic metrics can be calculated to characterize the local landscape surface. Unfortunately, a review of prior research suggests that an optimal set of variables have not been identified or recommended for slope failure susceptibility or occurrence mapping and that the optimal variable set is likely case-specific [17,20,21,43,58–60]. However, some measures have been shown to be of particular value, including measures of topographic slope, topographic wetness, and surface curvature [17,22,61–63]. Variable selection is further complicated by the reliance of many topographic variable calculations on local moving windows, which allows for defining neighborhoods using different window sizes and/or shapes [64]. Few studies have investigated the impact of widow size on variable importance and model performance for slope failure susceptibility or occurrence mapping specifically [17]. In our prior study, we found that variables using smaller windows were generally of greater importance in our slope failure occurrence model than the same variable calculated using a larger window [17]. However, this may not be the case for all landscapes. For a thorough review of terrain and geomorphometric variables for use in remote sensing and spatial predictive modeling, we recommend Franklin [65].

## 3. Methods

### 3.1. Study Area

Figure 1 shows the extent of the four MLRAs [66] explored in this study, while Table 1 provides the land areas and associated abbreviations used in this paper. The MLRAs include the Central Allegheny Plateau (CAP), Cumberland Plateau and Mountains (CPM), Eastern Allegheny Plateau and Mountains (EAPM), and Northern Appalachian Ridges and Valleys (NARV). Given that this project focuses on the state of West Virginia specifically, only the extents of these MLRAs within the state were considered. West Virginia, in general is characterized by a complex topography with a high degree of local relief. Elevations range from roughly 70 to 1480 m. Average winter temperatures are around 0 °C, while average summer temperatures are around 22 °C. Annual precipitation across the state is variable, which is partially attributed to topographic and rain shadow effects. The highest annual precipitation occurs on the western slopes of the high mountains within the EAPM MLRA, with totals as high as 1600 mm per year, while the lowest precipitation occurs in the NARV MLRA, with totals around 635 mm. The landscape is dominated by forests

but also includes areas of agriculture, pastureland, and development [67]. Although not extensively urbanized, as is common in many other regions in the eastern United States, the state has been heavily modified as a result of anthropogenic processes, land use/land cover change (LULCC), and resource extraction [27,67–69].

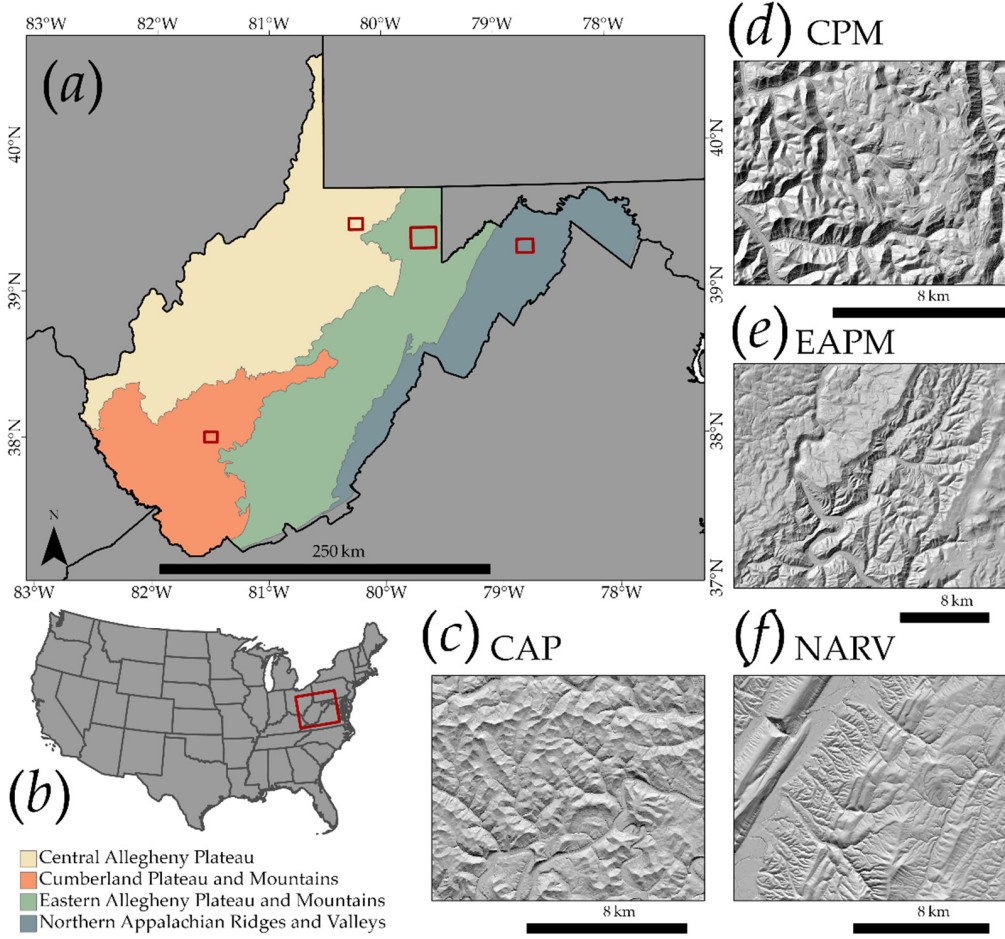

**Figure 1.** (**a**) Major Land Resource Areas (MLRAs) investigated and compared in the study. (**b**) Shows the extent of (**a**) in the contiguous United States. MLRA data are provided by the United States Department of Agriculture (USDA) [66]. (**c**) through (**f**) provide examples of terrain conditions, represented using hillshades, in the four MLRAs studied. (**c**) CAP = Central Allegheny Plateau, (**d**) CPM = Cumberland Plateau and Mountains, (**e**) EAPM = Eastern Allegheny Plateau and Mountains, (**f**) NARV = Northern Appalachian Ridges and Valleys.

**Table 1.** MLRA land areas, abbreviations used in this study, and number of mapped slope failure incidence points. Note that a statewide dataset of incidence points is not yet available since the LiDAR collection is not yet complete.

| MLRA | Abbreviation | Land Area in WV | Number of Slope Failures Mapped |
| --- | --- | --- | --- |
| Central Allegheny Plateau | CAP | 22,281 km$^2$ | 15,259 |
| Cumberland Plateau and Mountains | CPM | 11,644 km$^2$ | 12,533 |
| Eastern Allegheny Plateau and Mountains | EAPM | 18,071 km$^2$ | 12,438 |
| Northern Appalachian Ridges and Valleys | NARV | 10,320 km$^2$ | 1799 |

The CAP is a mature plateau with fine texture and a high degree of local relief dissected by a dendritic stream network. It is underlain by sedimentary rocks of mixed lithology but dominated by shale, mudstone, and sandstone, which are structurally flat to gently folding and date to the Pennsylvanian and Permian periods. Forest communities include oak-pine, oak-chestnut, cove hardwoods, and mixed mesophytic. The landscape has been modified by agriculture, development, and surface and underground coal mining [67,70]. The CPM has a similar topography as the CAP; however, relief tends to be more rugged due to several resistant geologic units. These topographic conditions have resulted in a concentration of development in the floodplains [67,70]. This region has also been heavily impacted by surface coal mining, including mountaintop removal practices, which have substantially altered slopes and landforms due to an inability to reclaim the landscape to a stable approximate original contour. Instead, mountaintops are flattened, forests are fragmented and replaced with dominantly herbaceous vegetation, and reclamation results in the filling of adjacent valleys with overburden materials [27,71–77]. It has been suggested that this region has experienced some of the highest global magnitudes of anthropogenic geomorphic alteration as a result of these mining practices, with substantial alterations to the physical landscape and drainage network [68,69,78,79].

The EAPM contains the highest elevations in the state, receives the highest levels of precipitation, and experiences the lowest annual winter temperatures. The lower elevation forests in this MLRA are similar to those in the CAP and CPM; however, higher elevations support northern hardwood and evergreen communities, including some native stands of red spruce. Geologic units are of mixed lithology and range in age from Devonian to Pennsylvanian, with some Mississippian limestones that result in karst valleys and associated landforms [67,70]. Lastly, the NARV is characterized as an eroded folded mountain belt, in which the topography is dominated by structural controls, including anticlines, synclines, and thrust faults. The stream network has a trellis pattern, and geologic units range in age from Precambrian to Devonian, with resistant sandstones and some limestones forming long, linear ridges and siltstones, shales, or less resistant limestones forming valleys. In comparison to the other MLRAs in the state, this region generally has the lowest elevations and receives the least amount of annual precipitation, especially on the east-facing slopes due to a rain shadow effect; as a result, drier forest communities dominate, including oak-pine and oak-hickory. This MLRA also includes a portion of the Great Valley, which is relatively flat and dominated by Cambrian and Ordovician limestone, dolomite, and shale with dominantly agricultural land use [67,70].

### 3.2. Landslide Inventory and Training Data Development

Models were trained and validated using reference data collected via manual interpretation of LiDAR-derived 1- or 2-m spatial resolution hillshades and slopeshades. The LiDAR data are made available via the USGS 3DEP program (https://www.usgs.gov/core-science-systems/ngp/3dep) (accessed on 2 May 2021) and the West Virginia GIS Technical Center (WVGISTC) and West Virginia View (http://data.wvgis.wvu.edu/elevation/) (accessed on 2 May 2021). Hillshades offer a visualization of the terrain surface, are derived from DTMs, and rely on the modeling of shadows cast over the landscape relative to an illuminating position, defined by azimuth and altitude parameters [80].

In contrast, a slopeshade is simply a topographic slope gradient raster grid in which shallower slopes are displayed using light shades and steeper slopes are displayed using darker shades. Slopeshades do not require defining the position of an illuminating source [17,27,80]. Due to the large spatial extent and number of features to be interpreted, and also as a result of the difficulty of accurately interpreting the areal extent of slope failure features, each identified feature was mapped as a point at the initiation location or head scarp. This process was completed by two trained analysts with supervision from a professional geomorphologist. Although we did not differentiate failure types in our probabilistic models, where possible, the analysts differentiated slides, debris flows, lateral spread, and multiple failures. Most features were categorized as slides.

Table 1 provides the number of mapped incident points available for this study. To date, a total of 42,029 features have been identified. Unfortunately, LiDAR data are not currently available for the entire state extent; however, data are currently being collected with a goal of a statewide LiDAR dataset being available by 2022. As a result, a statewide inventory is not yet possible. Figure 2 shows the spatial distribution of the inventory. All areas that have LiDAR data available have been interpreted, and only those areas were used in this study. Figure 3 shows some example incident points for all MLRAs included in this study.

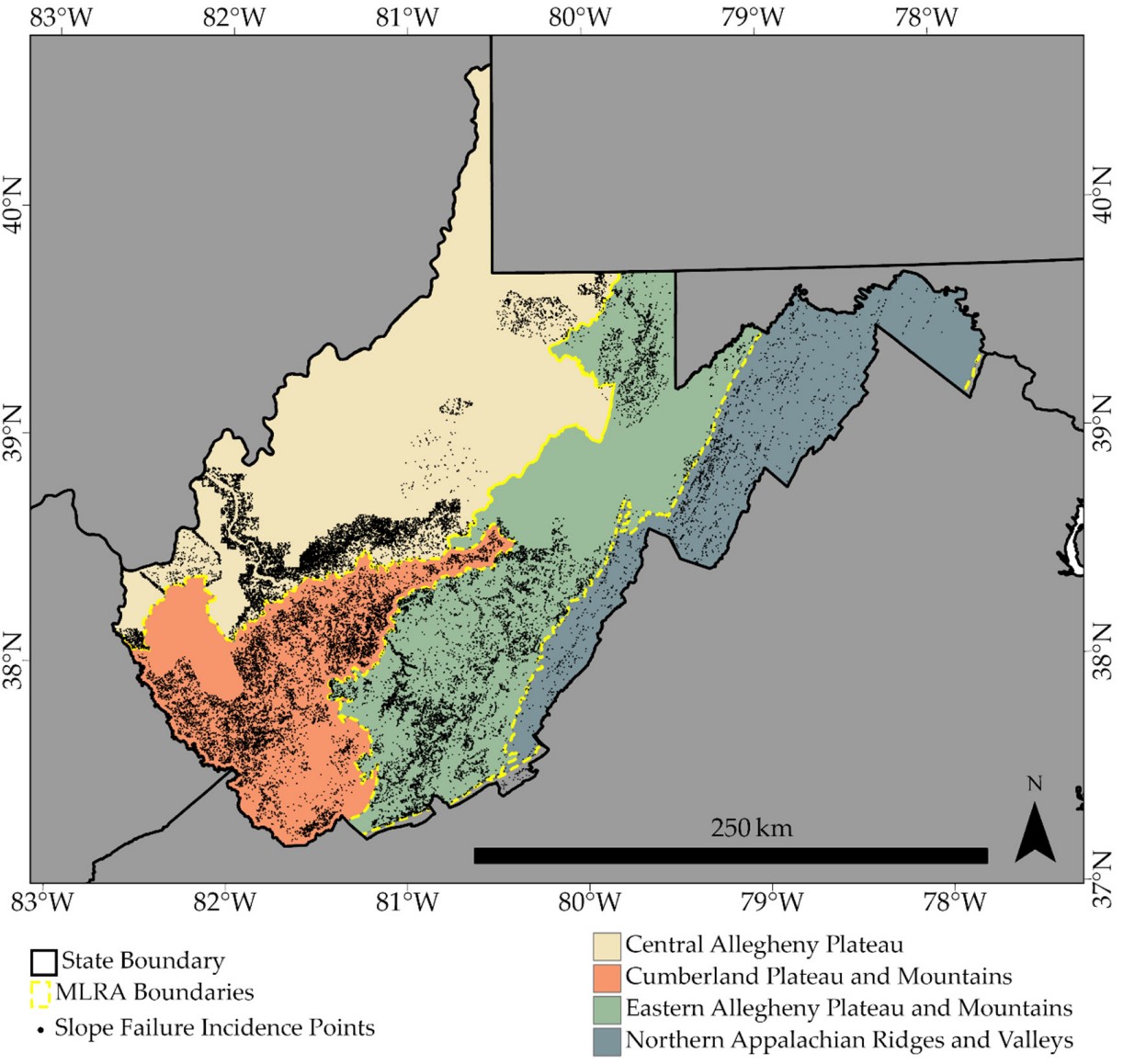

**Figure 2.** Distribution of slope failure incidence points relative to MLRA extents. All areas in West Virginia with LiDAR data available have been interpreted to generate the slope failure inventory.

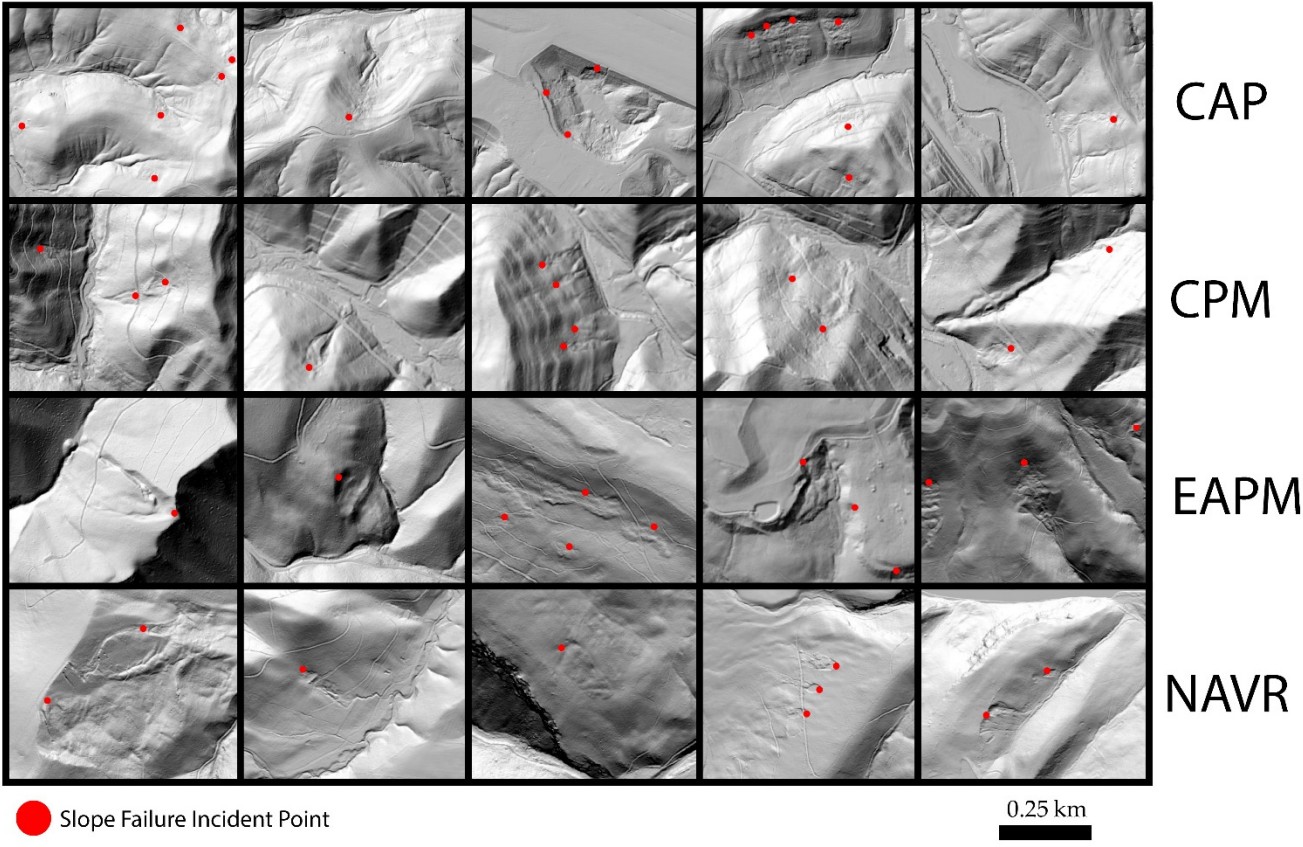

**Figure 3.** Example of slope failure incidence point mapped initiation locations or head scarps. Background is a hillshade derived from LiDAR data. Five examples have been provided for each MLRA investigated in this study.

Since the RF model requires both presence and absence data, we had to generate pseudo absence data to complement the slope failure inventory. To create these additional data, we generated random points throughout the MLRA extents. We then removed any points that (1) did not occur within areas in which LiDAR data are available, (2) occurred within 30 m of an inventoried landslide, or (3) occurred within the extent of or within 30 m of historic slope failure extents provided by the West Virginia Department of Transportation (WVDOT) and the West Virginia Geological and Economic Survey (WVGES). This same pseudo absence sampling method was used in our prior study [17]. Similar methods were used by Strager et al. [41], Maxwell et al. [28], and Maxwell et al. [81]. Given that the landslide inventories have been completed for all areas in the state where LiDAR data are available and that only these areas were included in this study, we argue that the likelihood of a pseudo absence sample occurring near a slope failure is unlikely.

### 3.3. Topographic Predictor Variables

Table 2 lists the terrain variables used in this study and provides the defined abbreviations and a brief description or equation. These terrain variables are identical to the variables used in our prior study [17]. In contrast to our prior study, in which we also included variables associated with distance from streams and roads, soil properties, and bedrock geology characteristics, in this study we only use terrain variables since (1) these variables can be consistently created for all areas where LiDAR data are available, allowing for ease of extrapolation of models to new geographic extents, and (2) our prior study suggested that including additional, non-terrain variables may not substantially improve model performance. All variables were calculated from a LiDAR-derived DTM with a 2-m spatial resolution, and a total of 14 different variables were produced. Several of these measures rely on moving windows to compare the center cell to its neighbors and derive

local terrain metrics. All variables that make use of moving windows were calculated using three different circular radii, 7, 11, and 21 cells, in order to characterize local patterns at different scales (Table 2). These scales were selected based on ridge-to-valley distances across the study area extents. This resulted in a total of 32 predictor variables.

**Table 2.** Description of terrain variables used in study. Abbreviations defined in this table are used throughout the paper.

| Variable | Abbreviation | Description | Window Radius (Cells) |
|---|---|---|---|
| Slope Gradient | Slp | Gradient or rate of maximum change in $Z$ as degrees of rise | 1 |
| Mean Slope Gradient | SlpMn | Slope averaged over a local window | 7, 11, 21 |
| Linear Aspect | LnAsp | Transform of topographic aspect to linear variable | 1 |
| Profile Curvature | PrC | Curvature parallel to direction of maximum slope | 7, 11, 21 |
| Plan Curvature | Plc | Curvature perpendicular to direction of maximum slope | 7, 11, 21 |
| Longitudinal Curvature | LnC | Profile curvature intersecting with the plane defined by the surface normal and maximum gradient direction | 7, 11, 21 |
| Cross-Sectional Curvature | CSC | Tangential curvature intersecting with the plane defined by the surface normal and a tangent to the contour-perpendicular to maximum gradient direction | 7, 11, 21 |
| Slope Position | SP | $Z$—Mean $Z$ | 7, 11, 21 |
| Topographic Roughness | TR | Square root of standard deviation of slope in local window | 7, 11, 21 |
| Topographic Dissection | TD | $\frac{Z - Min\ Z}{Max\ Z - Min\ Z}$ | 7, 11, 21 |
| Surface Area Ratio | SAR | $\frac{Cell\ Area}{\cosine(slope * \pi * 180)}$ | 1 |
| Surface Relief Ratio | SRR | $\frac{Mean\ Z - Min\ Z}{Max\ Z - Min\ Z}$ | 7, 11, 21 |
| Site Exposure Index | SEI | Measure of exposure based on slope and aspect | 1 |
| Heat Load Index | HLI | Measure of solar insolation based on slope, aspect, and latitude | 1 |

Slope gradient (Slp) [80] was calculated using the Slope Tool made available in the Spatial Analyst Extension of ArcGIS Pro [82]. The Geomorphometry & Gradient Metrics Toolbox [83] within ArcGIS Pro was used to calculate mean slope gradient (SlpMn) [80], linear aspect (LnAsp) [84], slope position (SP) [85], topographic roughness (TR) [86,87], topographic dissection (TD) [88], surface area ratio (SAR) [89], surface relief ratio (SRR) [59], site exposure index (SEI) [90], and heat load index (HLI) [91]. Profile (PrC), plan (PlC), longitudinal (LnC), and cross-sectional (CSC) curvatures [92,93] were calculated using the Morphometric Features Module in the open-source System for Automated Geoscientific Analysis (SAGA) software [94,95]. For a detailed introduction to geomorphometric variables derived from DTMs, we recommend Florinsky [96]. Ironside et al. [90] provide a review of such metrics in landscape ecology while Franklin [65] provides a review within remote sensing. Once all of the terrain variables were produced, the values at the pixel locations co-occurring with the presence and pseudo absence point data were extracted to generate data tables using the Extract Multi Values to Points Tool in ArcGIS Pro [82]. It should be noted that there is a correlation between the terrain predictor variables used. This was explored in our prior study [17] using Spearman's rho [97]. However, we documented that RF is robust to this complex feature space, as variable selection did not improve model performance.

### 3.4. Model Training and Prediction

Figure 4 provides a conceptualization of the data partitioning, training, validation, and inference processes used in this study. In order to reduce the influence of sample

size on the model comparisons, 1200 presence samples, which represent individual point or 2-m cell locations, were used to train each model. In order to validate the models, 500 non-overlapping presence samples were randomly withheld. So, each model was trained with the same number of presence samples and then evaluated using all of the validation sets, which were consistent for comparison and to assess model generalization to different MLRAs. As also implemented in our prior study [17] and in order to provide a wide variety of pseudo absence samples to characterize the landscape within the models, each MLRA separate model was trained five times, using 1200 training samples each time, and 1200 pseudo absence samples were selected using random sampling with a replacement. The trees from all models were then combined to create a final, single model for each MLRA. In an attempt to reduce spatial autocorrelation between the training and validation samples in each MLRA, each region was tessellated into 10,000-hectare contiguous hexagons. Random training and validation partitioning was conducted such that slope failure and pseudo absence samples within the same hexagon bin occurred in the same split.

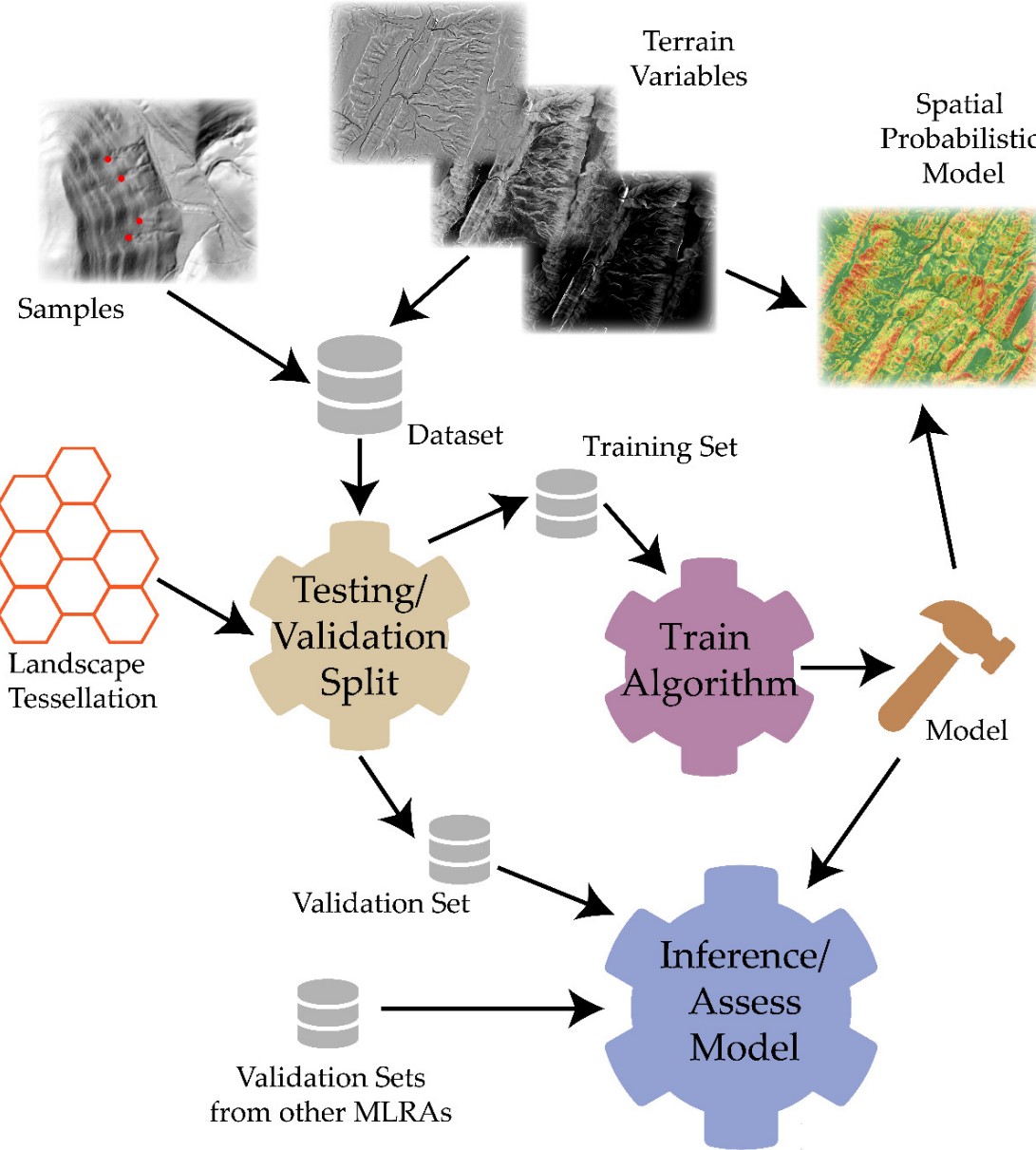

**Figure 4.** Conceptualization of modeling process used in this study including data partitioning, training, inference, and assessment. Database and hammer icons are from Font Awesome and made available under a CC by 4.0 license.

Models were trained using the randomForest package [98] in the R [99] data science environment. A total of 501 trees were used in each model, as this was adequate to stabilize the results. Once the five models were combined, this resulted in a total of 2505 trees in the final model for each MLRA. The number of predictor variables available for splitting at each node hyperparameter was optimized by testing 10 values using fivefold cross-validation and a grid search as implemented in the caret package [100]. Optimal performance was assessed using the AUC ROC metric. Each MLRA model was optimized separately.

### 3.5. Model Validation and Variable Importance

Models were evaluated using both binary, threshold-based metrics and also measures that do not rely on setting a classification probability threshold. For the threshold-based evaluation, samples with a predicted probability of occurrence in the presence, or slope failure, class of greater than or equal to 0.5 were classified to the positive case while those lower than 0.5 were mapped to the negative case. Table 3 describes the terminology used to define the binary assessment metrics used. True positive (TP) samples are those that are in the positive class and are correctly mapped as positive, in this case, slope failures, while false positives (FPs) are not in the positive class but are incorrectly mapped as positive. True negatives (TNs) are correctly mapped as negative, while false negatives (FNs) are mapped as negative when they are actually positive.

**Table 3.** Example binary confusion matrix and associated terminology. TP = True Positive, FP = False Positive, TN = True Negative, FN = False Negative.

| | | Reference Data | |
|---|---|---|---|
| | | **True** | **False** |
| Classification Result | True | TP | FP |
| | False | FN | TN |

Precision (Equation (1)) represents 1—commission error or the proportion of the samples that are correctly classified within the samples predicted to be positive. Recall or sensitivity (Equation (2)) represents 1—omission error or the proportion of the reference data for the positive class that is correctly classified. The F1 score (Equation (3)) is the harmonic mean of precision and recall, while specificity (Equation (4)) represents the proportion of negative reference samples that are correctly predicted. Overall accuracy (OA) (Equation (5)) represents the proportion of correctly classified features [101]. The Kappa statistic (Equation (6)) corrects OA for chance agreement [57]. All binary assessment metrics were calculated using the caret [100] package in R [99]. This package also allows for the calculation of 95% confidence intervals for OA based on a binomial distribution [100,102].

$$\text{Precision} = \frac{TP}{TP \ + \ FP} \tag{1}$$

$$\text{Recall or Sensitivity} = \frac{TP}{TP \ + \ FN} \tag{2}$$

$$\text{F1 Score} = \frac{2 \ \times \ \text{Precision} \ \times \ \text{Recall}}{\text{Precision} \ + \ \text{Recall}} \tag{3}$$

$$\text{Specificity} = \frac{TN}{TN \ + \ FP} \tag{4}$$

$$\text{Overall Accuracy} = \frac{TP + FP}{TP + TN + FP + FN} \tag{5}$$

$$\text{Kappa} = \frac{(\text{OA} - \text{Expected Agreement})}{(1 - \text{Expected Agreement})} \tag{6}$$

Receiver operating characteristic (ROC) curves and the associated area under the curve measure (AUC ROC) were used to provide an assessment that does not rely on a binary classification threshold. A ROC curve plots 1—specificity on the *x*-axis and sensitivity or recall on the *y*-axis at varying decision thresholds [101,103–105]. The AUC ROC measure is the area under the ROC curve and is equivalent to the probability that the classifier will rank a randomly chosen positive (true) record higher than a randomly chosen negative (false) record. ROC AUC is scaled from 0 to 1, with larger values indicating better model performance [101,103–106]. This analysis was undertaken using the pROC package [106] in R [99], which allows for the estimation of 95% confidence intervals for AUC ROC.

Since ROC curves and the associated AUC ROC metric rely on recall and specificity, which are both insensitive to an imbalance in regards to the number of positive and negative samples in the validation set, they can be misleading in cases where data imbalance should be taken into account, such as when the mapped classes make up very different proportions on the landscape. Specifically, reported ROC AUC can be overly optimistic in cases of severe class imbalance and/or when the class of interest makes up a small proportion of the investigated landscape [101,107]. In such cases, a precision-recall (P-R) curve may be more informative since it does incorporate precision, which is sensitive to class imbalance and quantifies the percentage of samples predicted to the positive class that were TPs. This curve plots sensitivity or recalls to the *x*-axis and precision to the *y*-axis. Similar to ROC, it is possible to generate an area under the curve (AUC PR) metric to obtain a single summary statistic [101,107,108]. This analysis was completed using the yardstick package [109] in R [99].

Variable importance measures produced by RF have been shown to be biased if predictor variables are highly correlated with one another [110,111]. As a result, and to offer a more rigorous evaluation and comparison of variable importance between the MLRAs, we used a measure of variable importance based upon conditional random forests that take into account correlation in the importance calculation as implemented in the R party package [94,95]. Importance ranges, means, and medians for each variable were created by running 10 separate assessments using different subsets of the training data.

## 4. Results

### 4.1. Model Performance within Same MLRA

Table 4 provides the validation statistics for each MLRA validation dataset predicted using the trained model from the same MLRA, while Figure 5 shows the distribution of predicted probabilities of slope failure occurrence for the slope failure and pseudo absence classes using kernel density functions. Generally, the slope failures and pseudo absence data are well separated with all OAs above 0.84, all Kappa statistics above 0.68, all F1 scores above 0.83, and all AUC ROC and AUC PR metrics above 0.90. However, the degree of overlap between the two distributions, summarized by precision (1—commission error for the slope failure class), recall (1—omission error for the slope failure class), and specificity (1—commission error for the pseudo absence class), vary between the four MLRAs. For example, the kernel density plots suggest that the separation of the classes is strongest for the EAPM and weakest for the CAP. This is further confirmed in Table 4, as OA, Kappa, F1 score, and AUC ROC for the EAPM are highest, and those for the CAP are lowest or tied with the CPM. Specifically, the EAPM has an OA of 0.890 and an AUC ROC of 0.957 while the CAP has an OA of 0.843 and an AUC ROC of 0.912.

Using the 0.50 probability threshold to classify the data, for each MLRA other than CAP recall is higher than precision, suggesting higher rates of FPs as opposed to FNs (i.e., higher commission error than omission error relative to the positive class). Specificity, which relates to rates of FPs, is lower than precision and recall for all MLRAs other than the CAP. Generally, the models tend to overestimate the extent of slope failure occurrence as opposed to overpredicting the extent of non-occurrence. It should be noted that assessing the models at different positive case probability thresholds will result in varying degrees of difference between precision, recall, and specificity [101].

**Table 4.** Accuracy assessment metrics for predicting each validation dataset using the model trained in the same MLRA. OA = Overall Accuracy, AUC ROC = Area Under the Receiver Operating Characteristic Curve, AUC PR = Area Under the Precision-Recall Curve.

| MLRA | OA | Kappa | Precision | Recall | Specificity | F1 Score | AUC ROC | AUC PR |
|------|------|------|------|------|------|------|------|------|
| CAP | 0.843 | 0.686 | 0.870 | 0.806 | 0.880 | 0.837 | 0.912 | 0.911 |
| CPM | 0.843 | 0.686 | 0.836 | 0.854 | 0.832 | 0.845 | 0.914 | 0.905 |
| EAPM | 0.890 | 0.780 | 0.864 | 0.926 | 0.854 | 0.894 | 0.957 | 0.949 |
| NAVR | 0.879 | 0.758 | 0.858 | 0.908 | 0.850 | 0.882 | 0.952 | 0.952 |

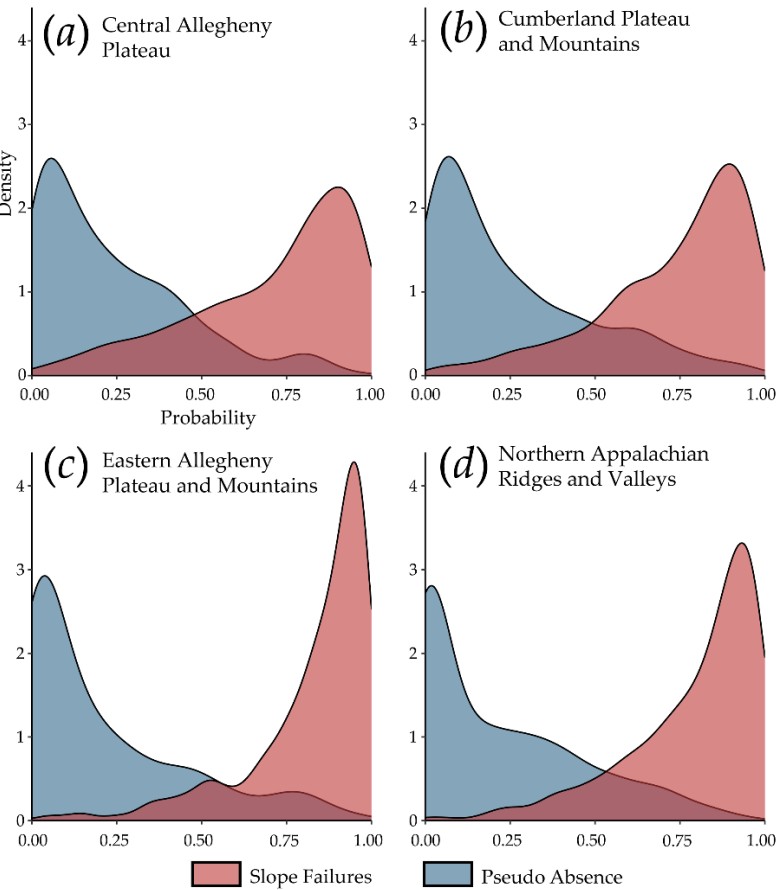

**Figure 5.** Probability distribution of withheld slope failure and pseudo absence validation samples for each region. Probability relates to the likelihood of slope failure occurrence. (**a**) Central Allegheny Plateau, (**b**) Cumberland Plateau and Mountains, (**c**) Eastern Allegheny Plateau and Mountains, (**d**) Northern Appalachian Ridges and Valleys.

*4.2. Model Generalization to Different MLRAs*

Table 5 provides the summary statistics for predicting each validation dataset with all four models to provide an assessment of model generalization to new geographic extents. The rows in which the validation set and model are the same are identical to the data provided in Table 4. For all validation datasets, the model trained using data from that same MLRA provided the highest reported OA, Kappa statistic, F1 Score, AUC ROC, and AUC PR. However, the model from the same MLRA as the validation dataset did not always provide the highest precision, recall, and/or specificity. Generally, this suggests that models extrapolated to new geographic extents tend to either overpredict or underpredict the land area of slope failure occurrence, whereas they offer a better balance between omission and commission error within the region in which they were trained.

**Table 5.** Accuracy assessment metrics for predicting each validation dataset using each model. The highest metrics for each validation dataset are shaded gray.

| Validation Set | Model | OA | Kappa | Precision | Recall | Specificity | F1 Score | AUC ROC | AUC PR |
|---|---|---|---|---|---|---|---|---|---|
| CAP | CAP | 0.843 | 0.686 | 0.870 | 0.806 | 0.880 | 0.837 | 0.912 | 0.911 |
| CAP | CPM | 0.641 | 0.282 | 0.851 | 0.342 | 0.940 | 0.488 | 0.847 | 0.812 |
| CAP | EAPM | 0.795 | 0.590 | 0.814 | 0.764 | 0.826 | 0.788 | 0.860 | 0.848 |
| CAP | NAVR | 0.751 | 0.502 | 0.727 | 0.804 | 0.698 | 0.764 | 0.821 | 0.809 |
| CPM | CAP | 0.734 | 0.468 | 0.750 | 0.702 | 0.766 | 0.725 | 0.800 | 0.749 |
| CPM | CPM | 0.843 | 0.686 | 0.836 | 0.854 | 0.832 | 0.845 | 0.914 | 0.905 |
| CPM | EAPM | 0.734 | 0.468 | 0.662 | 0.958 | 0.510 | 0.783 | 0.869 | 0.840 |
| CPM | NAVR | 0.662 | 0.324 | 0.604 | 0.944 | 0.380 | 0.736 | 0.769 | 0.739 |
| EAPM | CAP | 0.836 | 0.672 | 0.891 | 0.766 | 0.906 | 0.824 | 0.918 | 0.901 |
| EAPM | CPM | 0.796 | 0.592 | 0.925 | 0.644 | 0.948 | 0.759 | 0.935 | 0.933 |
| EAPM | EAPM | 0.890 | 0.780 | 0.864 | 0.926 | 0.854 | 0.894 | 0.957 | 0.949 |
| EAPM | NAVR | 0.849 | 0.698 | 0.794 | 0.942 | 0.756 | 0.862 | 0.931 | 0.916 |
| NAVR | CAP | 0.758 | 0.516 | 0.919 | 0.566 | 0.950 | 0.700 | 0.897 | 0.890 |
| NAVR | CPM | 0.677 | 0.354 | 0.959 | 0.370 | 0.984 | 0.534 | 0.893 | 0.887 |
| NAVR | EAPM | 0.830 | 0.660 | 0.886 | 0.758 | 0.902 | 0.817 | 0.928 | 0.927 |
| NAVR | NAVR | 0.879 | 0.758 | 0.858 | 0.908 | 0.850 | 0.882 | 0.952 | 0.952 |

The OA, Kappa statistic, AUC ROC, and AUC PR metrics reported in Table 5 are further visualized in Figure 6, which also provides the estimated 95% confidence intervals for OA and AUC ROC. Although the model trained within the same MLRA as the validation dataset provided the best performance for all four MLRAs, as measured with these four assessment metrics, the confidence intervals do overlap between some of the models, and some models offer accuracies that are only slightly lower in comparison to the model trained within the MLRA represented by the validation dataset. For example, the NARV model performed well when predicting to the EAPM validation dataset (OA = 0.849, Kappa statistic = 0.698, F1 score = 0.862, AUC ROC = 0.931, and AUC PR = 0.916) in comparison to the EAPM model (OA = 0.890, Kappa statistic = 0.780, F1 score = 0.894, AUC ROC = 0.957, and AUC PR = 0.949). MLRAs showed variable levels of disparity between the four models. For example, the CAP MLRA was generally poorly predicted by all other models whereas the EAPM was generally predicted well by all models. The CPM was predicted well by the CPM and EAPM models, but more poorly by the CAP and NARV models.

Similar patterns are evident in the ROC curves (Figure 7) and P-R curves (Figure 8), and the two assessment methods reinforce each other in regards to model performance and model variability for predicting the same MLRA validation datasets. All four models performed more similarly for the EAPM in comparison to the other three MLRAs. Generally, these curves, and the associated AUC metrics, support the findings from the binary assessment metrics and extend the results across the spectrum of binary classification decision thresholds, as the models trained and applied to the same MLRA generally resulted in higher recall at any given specificity or higher precision at any given recall.

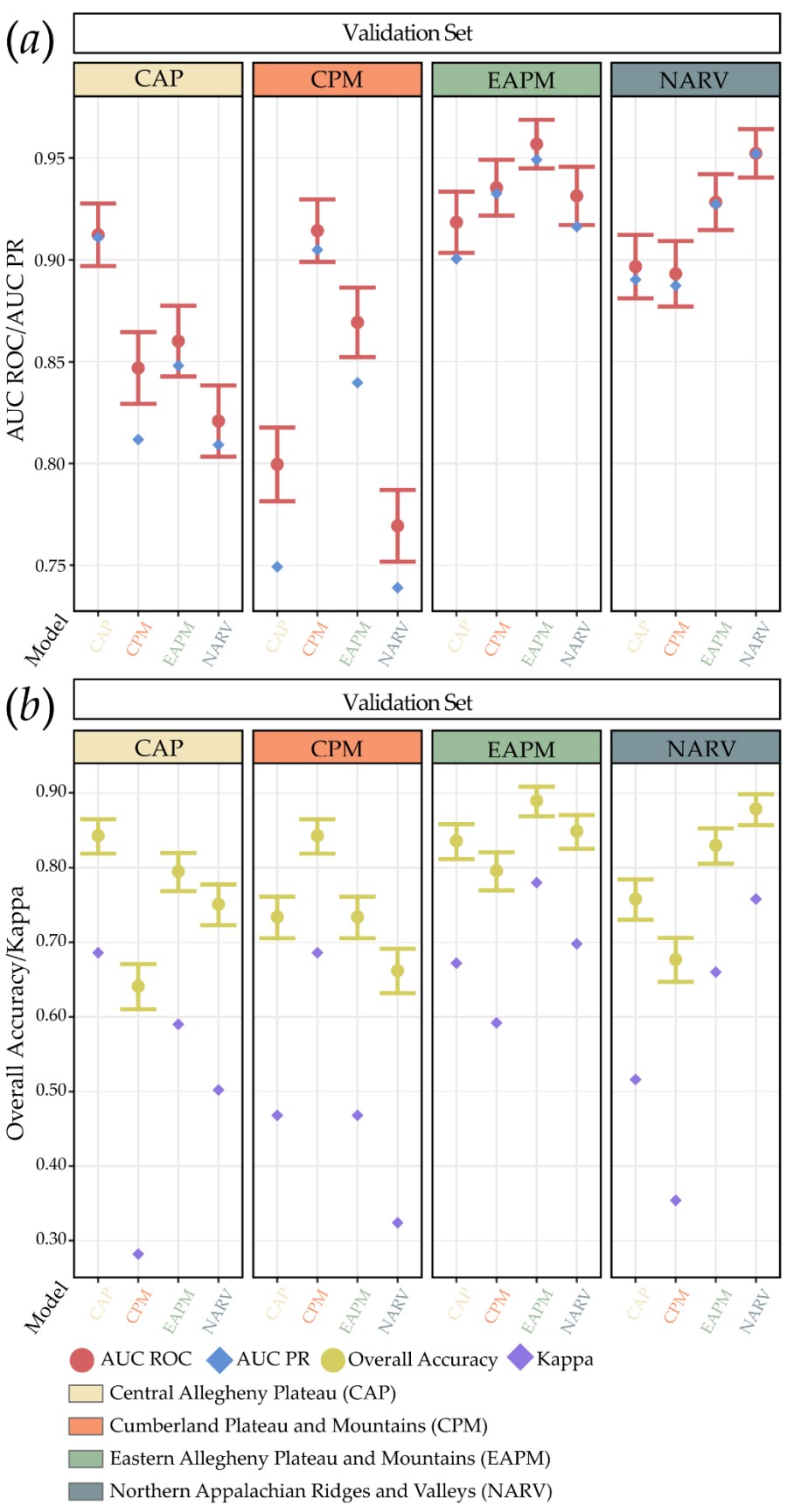

**Figure 6.** Overall Accuracy, Kappa, AUC ROC, and AUC PR for predicting each validation dataset using each model. Bars represent a 95% confidence interval for overall accuracy and AUC ROC. (**a**) AUC ROC and AUC PR. (**b**) Overall accuracy and Kappa.

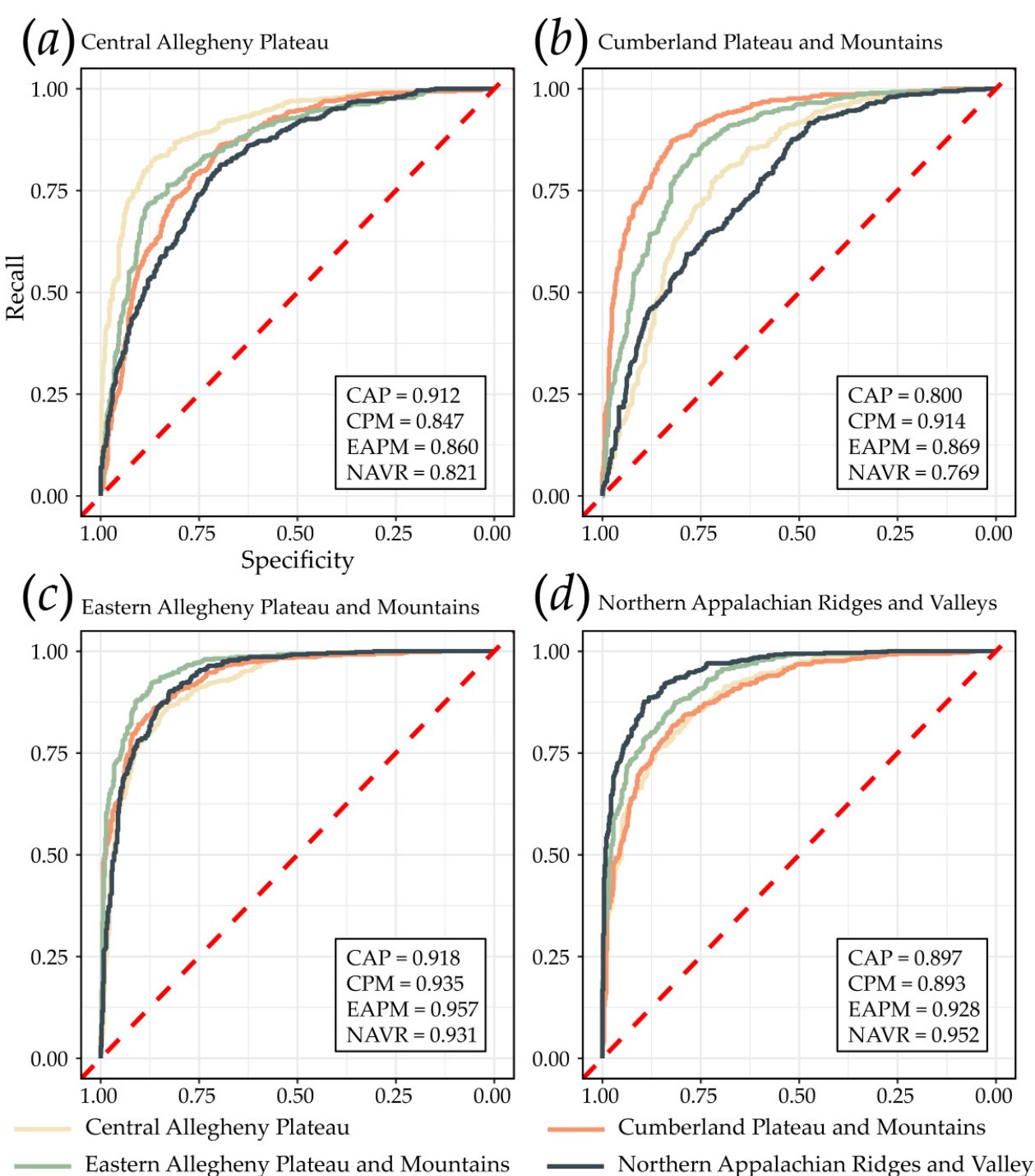

**Figure 7.** ROC curves for predicting each MLRA validation dataset using each model. Each plot represents the MLRA area predicted, while each curve represents the model used to make the predictions. Inset boxes provide the associated AUC for each curve. (**a**) Central Allegheny Plateau, (**b**) Cumberland Plateau and Mountains, (**c**) Eastern Allegheny Plateau and Mountains, (**d**) Northern Appalachian Ridges and Valleys.

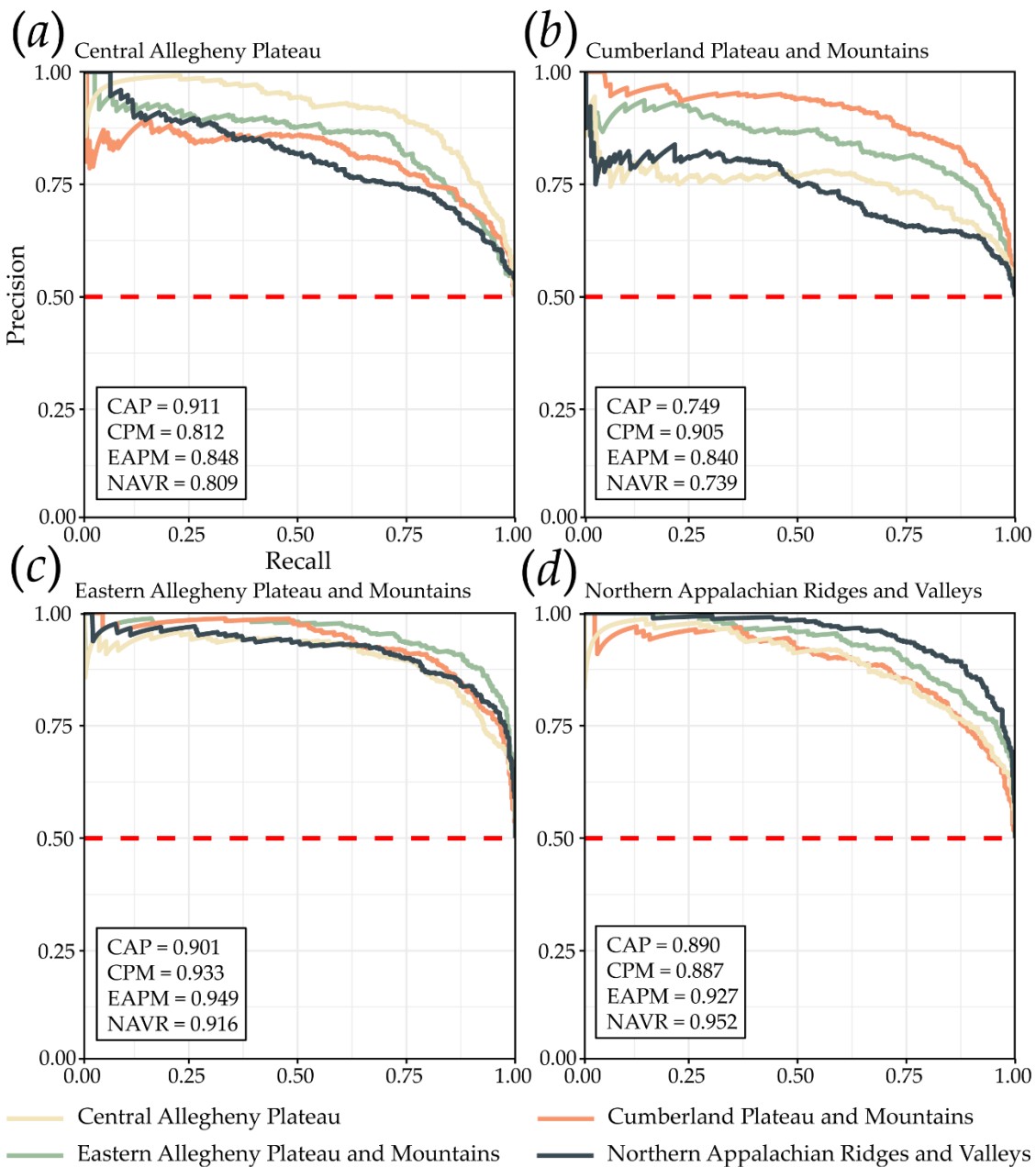

**Figure 8.** P-R curves for predicting each MLRA validation dataset using each model. Each plot represents the MLRA area predicted, while each curve represents the model used to make the predictions. Inset boxes provide the associated AUC for each curve. (**a**) Central Allegheny Plateau, (**b**) Cumberland Plateau and Mountains, (**c**) Eastern Allegheny Plateau and Mountains, (**d**) Northern Appalachian Ridges and Valleys.

### 4.3. Comparison of Variable Importance Between MLRAs

Figure 9 compares variable importance for each model calculated using the conditional variable importance methods after Strobl et al. [111]. Generally, the results suggest variable importance does vary between the regions. However, some key variables are consistently highlighted as important for the predictions, including Slp, SAR, CSC, and PlC. These findings generally support those from our prior study [17], where Slp, SAR, CSC, and PlC were also found to be important. Slp was the most important variable on average for all the models, except the CPM, where SP calculated with a radius of 7 m was found to be most important followed by Slp. Some variables were also consistently shown to be of low importance, including HLI and LnAsp. In contrast to our prior study [17], the impact

of window size was variable between the MLRAs. Lastly, the EAPM and NARV MLRAs, which had the highest reported OA, Kappa, AUC ROC, and AUC PR, had a small subset of variables that were found to be of high importance while all remaining variables were of low importance (Figure 9c,d).

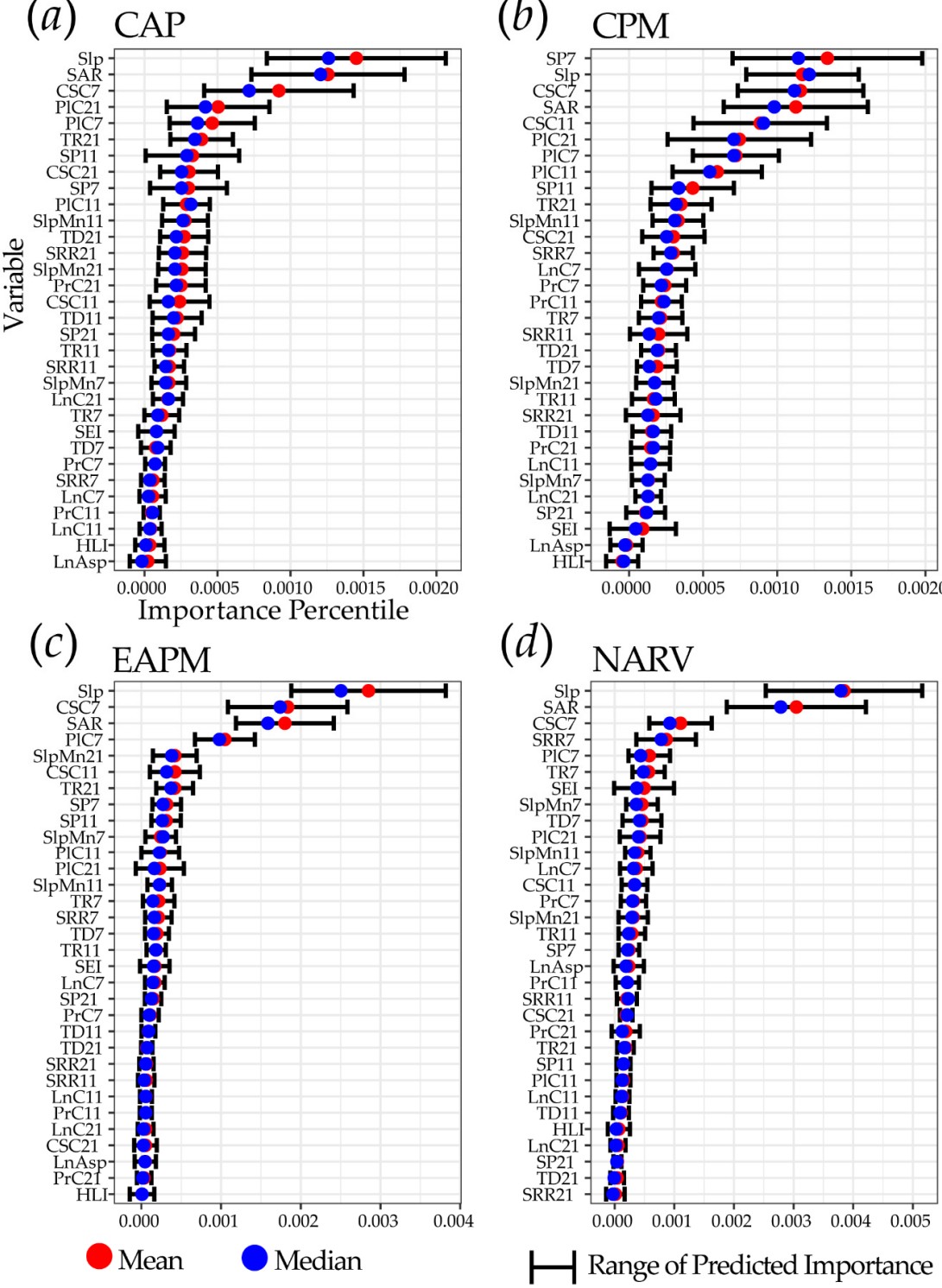

**Figure 9.** Conditional variable importance estimates for each separate MLRA. Bars represent the range of importance calculated from 10 replicates of the analysis using different subsets of the training samples. (**a**) Central Allegheny Plateau, (**b**) Cumberland Plateau and Mountains, (**c**) Eastern Allegheny Plateau and Mountains, (**d**) Northern Appalachian Ridges and Valleys.

## 5. Discussion

In support of the findings from our prior study [17], this study generally suggests that the RF ML algorithm is capable of generating accurate slope failure occurrence probabilistic models from point-based inventories and digital terrain variables derived from high spatial resolution and detailed LiDAR data without the need to include additional, non-terrain variables. Expanding upon the prior study, this research suggests that models trained in different landscapes generally do not provide comparable accuracies to models applied to new data in the same landscape. Or, these ML models did not generalize well to new landscapes.

Furthermore, the disparity between models was inconsistent between MLRAs. For example, model accuracies were more similar between all four models when applied to the EAPM than the CAP. Some models provide higher recall, precision, or sensitivity for predicting to new study area extents than the models trained in those extents. However, OA, the Kappa statistic, F1 score, AUC ROC, and AUC PR were always highest for the model trained in the same MLRA as the validation data. This suggests that models will underpredict or overpredict the occurrence of slope failures in new landscapes in comparison to models trained in that landscape.

Despite the poorer performance when applied to new landscapes, several variables were found to be consistently important between models, including Slp, SAR, CSC, and PlC. The EAPM and NARV MLRAs were predicted with the highest accuracies and also had a smaller subset of the variables that were predicted to be of higher importance than all other variables provided. This could be a result of a more distinctive slope failure signature that can be described with a smaller number of variables to differentiate these features from the rest of the landscape. However, it was not possible to test this assertion.

These findings are similar to the findings of Maxwell et al. [28]. They investigated the probabilistic prediction of palustrine wetland occurrence using digital terrain variables and the generalization of these models to new physiographic regions. Their results suggested that models trained in different physiographic regions did not perform as well as models trained within the region of interest.

Although this study quantified reduced model performance when generalized to new landscapes, the reasons for this reduction cannot be determined. For example, it is possible that reduced performance relates to differences in landscape characteristics between regions and/or differences in the characteristics or causes of slope failures. For example, visual inspection of the slope failure inventory within the CPM suggests that many failures occur near mine or reclaimed mine sites, an anthropogenic driver of landscape change and potential slope instability that is not abundant in the other MLRAs studied. There are also key differences in underlying geology. For example, the NARV is geomorphically characterized by long linear ridges and valleys resulting from folding and mountain building and subsequent erosion. In contrast, the landscapes of the CAP and CPM are less controlled by geologic structure, since the rock units are flat to gently folding [67,70].

In order to address the specific question of model generalization to new landscapes without introducing bias into the comparison, we used the same sample size (1200 randomly selected slope failure locations and 1200 pseudo absence samples) to minimize the impact of the number of training samples. We also created non-overlapping validation sets and employed a hexagon tessellation for geographic stratification to minimize the impact of model overfitting. Assuming overfitting was minimized by our training and sampling methods, we argue that lack of generalization to new study area extents may be explained by a difference in landscape characteristics and/or slope failure characteristics between the regions. Or, new, withheld validation samples within an MLRA are more similar to training samples within that MLRA than samples from a different MLRA, resulting in differences in model fit.

Future studies should investigate generalization to more disparate landscapes. Although our regions were different, they all occurred within West Virginia and were adjacent. It would be interesting to perform similar analyses using very different landscapes,

such as young mountain belts, eroded mountains, regions impacted by recent glaciation, paraglacial landscapes, and regions underlain by very different lithologies (e.g., sedimentary vs. metamorphic units). It is also of interest to explore DL methods for generating slope failure occurrence and susceptibility predictions and to assess model generalization. Recent research has suggested that DL methods can generalize better than traditional ML methods [31,32]. However, such assumptions have not been widely explored for geomorphic mapping and probabilistic modeling specifically.

Practically, this research highlights the value of large landslide inventories covering different landscapes, such as the US Landslide Inventory [112], for training and validating region-specific models. It also highlights the value of high spatial resolution and detailed digital terrain data, such as the data made available by USGS 3DEP [23,24], since generating large training and validation datasets and being able to describe key terrain characteristics are necessary for generating spatially detailed probabilistic models. If models trained in one area must be applied to new areas due to a lack of training data, it is important that some form of validation be applied since the accuracy obtained in new landscapes will likely not be characterized by a validation in the original landscape. This study will be informative for practitioners planning and undertaking operational, large area hazard risk assessments, especially if the area of interest has variable terrain and geomorphic characteristics. Within West Virginia, our models have already proved useful for risk assessment and management activities, and we hope that this work will inform the application of similar methods in new states or regions.

## 6. Conclusions

This study documents reduced performance of slope failure models when extrapolated to new geographic extents, as defined by MLRAs. Models trained in disparate geographic extents were not able to obtain the level of accuracy provided by a model trained and validated to the same geographic extent. Errors of commission or omission were sometimes reduced when predicting using a model from a different MLRA. However, the model trained in the same region as the validation data provided the best balance between omission and commission errors, as measured with the F1 score. This study highlights the importance of training models relative to specific geographic extents. It should not be assumed that models will generalize well, which highlights the value of region-specific datasets for training and validation.

**Author Contributions:** Conceptualization, Aaron E. Maxwell; methodology, Aaron E. Maxwell.; validation, J. Steven Kite, Shannon M. Maynard and Caleb M. Malay; formal analysis, Aaron E. Maxwell and Caleb M. Malay; writing—original draft preparation, Aaron E. Maxwell; writing—review and editing, Aaron E. Maxwell, Maneesh Sharma, J. Steven Kite, Kurt A. Donaldson, Shannon M. Maynard and Caleb M. Malay; data curation, Maneesh Sharma, Kurt A. Donaldson, Shannon M. Maynard and Caleb M. Malay; supervision, Maneesh Sharma, J. Steven Kite and Kurt A. Donaldson; project administration, Maneesh Sharma and Kurt A. Donaldson; funding acquisition, Maneesh Sharma and Kurt A. Donaldson. All authors have read and agreed to the published version of the manuscript.

**Funding:** Funding for this research has been provided by FEMA (FEMA-4273-DR-WV-0031). The performance period for the project is 20 June 2018 to 4 June 2021.

**Data Availability Statement:** Example data and code associated with this study are made available on GitHub (https://github.com/maxwell-geospatial/slopefailure_prob_models) (accessed on 2 May 2021). The full datasets can be made available upon request by contacting the corresponding author.

**Acknowledgments:** We would like to acknowledge the Federal Emergency Management Agency (FEMA) and the West Virginia Division of Emergency Management (WVDEM) for funding this risk assessment project under the Hazard Mitigation Grant Program. We would like to acknowledge initial encouragement and support for a landslide risk assessment study for West Virginia by State Hazard Mitigation Officer Brian Penix. Ray Perry, Floodplain Manager, is acknowledged for help in identifying landslide locations in Logan County, WV and Elizabeth Hanwell and Matthew Bell, student interns at the WVGISTC, for providing valuable help during various phases of this study.

We would also like to thank the three anonymous reviewers whose suggestions and comments strengthened the work.

**Conflicts of Interest:** The authors declare no conflict of interest.

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
