# Peer review of "Assessing the Generalization of Machine Learning-Based Slope Failure Prediction to New Geographic Extents"

_ijgi, doi:10.3390/ijgi10050293_

Round 1

Reviewer 1 Report

1. Line 100-101 is unclear please rewrite more specifically 2. Authors should add more literature reviews regarding RF and landslide susceptibility 3. Authors should add DEM in Figure 1 to support their statement from line 189 to 205 4. In line 205, land use and land cover or land use/land cover. I suggest you to revise into land use/land cover 5. Why authors used landslide incident points instead of polygons? How shape and size is represented in the model. 6. Figure 3 have poor resolution 7. Authors should elaborate how they achieved optimum ntree and mtry. Are they used grid search? If yes then what is the criteria to achieve optimum tuning parameters 8. What are the multi-collinearity statistics between the selected independent variables? Since all these factors are DEM derivatives, I think there is a chance of multi- collinearity. How authors will address that? Authors should do a sensitivity analysis to find out the effect of cross-correlation. 9. Even though larger spatial extend and huge dataset is available why authors avoided geographical cross-validation. I suggest to the authors to do geographical cross- validation. 10. Authors should include AUC-ROC values in the Figure 6 itself. 11. Beyond model prediction and generalization, how this study benefits in the study area? Authors should add a separate section for that. 12. Where is the details of previous landslide events? 13. Where is the maps of independent variables?

Reviewer 2 Report

The paper is devoted to the study of an efficiency of slope failure predictions models developed for the specific geographic areas when applied to other ones. The main conclusion of the paper is that models training is of high importance in every specific geographic extent.

Low efficiency of the models is not surprising. Among probable reasons is a lack of the features used for models training – the only source of slopes failures signatures construction are topographic data. Authors make correct supposition about usefulness of some other types of information like as geomorphological description of the landscape.

The paper is well-organized and written in plain language. Among few insignificant remarks is an undefined term “sample” in line 328. How the sample size relates to DTM resolution cell from line 303?

Reviewer 3 Report

This paper presents the results of extrapolating the slope failure models to new geographic extents. The data set is selected in West Virginia and classified into four different types of land areas. The research overall is complete, and the discrepancy is analyzed. There are few comments that can be considered to improve the manuscript. Please see below.

Specific comments:

  1. The introduction section is wordy.
  2. Please use a flow chart to indicate the training process as this is the core of this paper.

Other detailed comments

  1. L36: Reference overkill.
  2. L38: Same as above.
  3. L79-80: reference.
  4. L286: why do you think 30 m is an approximate reference value?
  5. L372: please type in the equation properly.
  6. Figure 8: make sure the label size is consistent.
